# The Association of Prenatal Antibiotic Use with Attention Deficit and Autism Spectrum Disorders: A Nationwide Cohort Study

**DOI:** 10.3390/children10071128

**Published:** 2023-06-29

**Authors:** Yu-Chun Lin, Ching-Heng Lin, Ming-Chih Lin

**Affiliations:** 1Children’s Medical Center, Taichung Veterans General Hospital, Taichung 407, Taiwan; ativaniin@gmail.com; 2Department of Medical Research, Taichung Veterans General Hospital, Taichung 407, Taiwan; epid@vghtc.gov.tw; 3Department of Post-Baccalaureate Medicine, College of Medicine, National Chung Hsing University, Taichung 402, Taiwan; 4School of Medicine, National Yang Ming Chiao Tung University, Taipei 112, Taiwan; 5Department of Food and Nutrition, Providence University, Taichung 433, Taiwan; 6School of Medicine, Chung Shan Medical University, Taichung 402, Taiwan

**Keywords:** attention deficit hyperactivity disorder, autism spectrum disorder, prenatal exposure, antibiotics

## Abstract

(1) Background: Attention deficit hyperactivity disorder (ADHD) and autism spectrum disorder (ASD) are common cognitive and behavioral disorders. Antibiotics are widely used in pregnant women and their newborns. The objective of this study was to examine the potential association between prenatal exposure to antibiotics and the risk of ADHD and ASD in childhood from a nationwide perspective. (2) Methods: The Taiwan National Health Insurance Research Database (NHIRD) was used as the primary data source. This nationwide cohort study included only first-time pregnancies. A total of 906,942 infants were enrolled. All infants were followed up for at least 6 years. The Cox regression model was applied for covariate control. (3) Results: Prenatal exposure to antibiotics was found to significantly increase the cumulative incidence of ADHD while having only a borderline effect on the cumulative incidence of ASD. Exposure to antibiotics during any of the three different gestational age ranges significantly increased the cumulative risk. However, only exposure after 34 weeks of gestation had a significant impact on the occurrence of ASD. The study also revealed a dose-dependent effect on the occurrence of ADHD but no effect on the occurrence of ASD. (4) Conclusions: This study suggests that prenatal exposure to antibiotics may increase the risk of developing ADHD and ASD later in life.

## 1. Introduction

Attention deficit hyperactivity disorder (ADHD) and autism spectrum disorder (ASD) are common cognitive and behavioral disorders that exhibit an increasing prevalence in childhood worldwide. Epidemiological and twin studies suggest that ADHD frequently co-occurs with and shares genetic susceptibility with autism spectrum disorder (ASD) and ASD-related traits [1,2]. Besides genetic factors, several established risk factors for these disorders include heritability, preterm birth, maternal or neonatal illness, and exposure to environmental toxins, such as heavy metals, cigarettes, and alcohol [1,2,3,4,5,6]. Moreover, nutritional status during preconception and the prenatal period is one of the modifiable factors, and the intake of folic acid and multivitamins has been found to be inversely associated with the development of these disorders [7]. Essential elements may also act on cellular, molecular, and biochemical roles, affecting the plasticity of neural structures [8]. Moreover, maternal use of certain medications, such as anti-epileptic drugs, e.g., valproate, during pregnancy can also increase the risk of ADHD in offspring [9,10].

Antibiotics are widely used medications for infection control that have been shown to alter the gut microbiomes of pregnant women and their newborns. This exposure can lead to changes in the composition and diversity of the intestinal flora, which may impact brain–gut-microbiome interactions [11,12]. The intestinal microbiome also regulates the environment of the gastrointestinal tract by means of the miroorganisms or their metabolites, and it may affect psychological status by the activation of the autonomic nervous system through the synthesis of varous neurotransmitters [13]. Recent research has suggested that alterations in the gut microbiome may be implicated not only in gastrointestinal disease but also in neurologic and psychiatric disorders [14]. Thus, prenatal antibiotic use might have an impact on the pathogenesis of ADHD and ASD later in life through brain–gut-microbiome interactions.

Despite these findings, the relationship between prenatal or intrapartum antibiotic use, the local or systemic microbiome, and the development of ADHD and ASD are still not well understood. Specifically, the association between maternal use of antibiotics during pregnancy and the later development of ADHD and ASD in offspring has not been extensively studied. The objective of this study was to examine the potential association between prenatal exposure to antibiotics and the risk of ADHD and ASD in childhood from a nationwide perspective.

## 2. Materials and Methods

### 2.1. Study Design and Data Source

This was a nationwide, population-based cohort study that used the Taiwan National Health Insurance Research Database (NHIRD) as the primary data source. Taiwan’s National Health Insurance (NHI) system was established in 1995 as a single-payer program with mandatory enrollment, covering nearly the entire population of Taiwan, estimated to be around 23.5 million individuals, with a current coverage rate of 99.99%. In 2002, the NHIRD was created specifically for research purposes and included all claims data from the NHI [15,16,17]. In 2015, the Health and Welfare Data Center (HWDC) of Taiwan’s Ministry of Health and Welfare (MOHW) further integrated the NHIRD with other health-related databases [18]. The pairing of mothers and children was accomplished by linking the NHIRD with the Taiwan Maternal and Child Health Database (MCHD) of Taiwan’s Health Promotion Administration (HPA). The primary data analyzed in this study were acquired from ambulatory care expenditures by visit (CD) files and inpatient expenditures by admission (DD) files from the NHIRD. Antibiotic exposure records were obtained from both inpatient order (DO) and outpatient order (OO) files. To safeguard privacy and ensure database reliability, Taiwan’s Ministry of Health and Welfare (MOHW) mandates on-site analysis by investigators. During the study period, diagnoses in the NHIRD were coded in the International Classification of Diseases, Ninth Revision, Clinical Modification (ICD-9-CM) format. The study protocol was reviewed and approved by the institutional review board of the Taichung Veterans General Hospital. The review board also waived the need for informed consent for this study.

### 2.2. Study Population

This nationwide cohort study included only first-time pregnancies from 2004 to 2010. Infants of multiple deliveries, preterm delivery, and those who died before reaching 5 years old were excluded.

### 2.3. Exposure to Antenatal Antibiotics

In this study, antenatal antibiotic exposure was defined as a mother who received medication with an Anatomical Therapeutic Chemical (ATC) code of J01A, J01B, J01C, J01D, J01E, J01F, J01G, J01M, J01R, or J01X during pregnancy. We also recorded the timing of the prescription, including the first, second, and third trimesters, as well as the cumulative number of prescriptions received.

### 2.4. Outcome Measurement

ADHD was diagnosed based on ICD-9 codes 314.0 (attention deficit disorder of childhood), 314.00 (attention deficit disorder without mention of hyperactivity), and 314.01 (attention deficit disorder with hyperactivity). ASD was diagnosed based on ICD-9 codes 299.0 (autistic disorder), 299.00 (autistic disorder, current or active state), and 299.01 (autistic disorder, residual state), as well as 315.9 (unspecified developmental delay). Children who visited the outpatient department three times or more, or who were admitted at least once with a diagnosis, were considered to have ASD or ADHD.

### 2.5. Covariates

We also gathered data on potential confounding variables, including maternal age, mode of delivery, maternal comorbidities, maternal allergic diseases, pregnancy-related complications, and the gender of the infant.

### 2.6. Statistical Analysis

The data were analyzed using SAS statistical software version 9.4 (SAS Institute, Cary, NC, USA). Descriptive statistics were reported as means with standard deviations, or as frequencies and percentages. Continuous variables were compared using an independent *t*-test, while categorical data were compared using the Pearson’s chi-square test. The cumulative incidences of ADHD and ASD were compared between groups using the Kaplan–Meier method. Hazard ratios for antibiotic prescription timing and frequency were calculated using the Cox regression model, with adjustment for potential confounders. Statistical significance was set at a *p*-value of less than 0.05.

## 3. Results

A total of 906,942 infants were enrolled in the study cohort after applying the exclusion criteria (Figure 1). This nationwide cohort study included only first-time pregnancies from 2004 to 2010. The cohort was followed up until the end of 2016. All children were followed up for at least 6 years. This is a claim data study. Only children with claim data during the study period were included. Children who died during the study period were excluded. The follow-up time was 6 to 12 years. The mean follow-up period was around 9 years.

Among the 906,942 infants enrolled, 53.4% had a history of antenatal antibiotic exposure. The demographic data are presented in Table 1. Advanced or younger maternal age was positively associated with increased antenatal antibiotic use, as was delivery by cesarean section. Maternal comorbidities, such as diabetes, hypertension, hyperlipidemia, and mental health issues, were also found to be associated with an increased likelihood of antenatal antibiotic exposure. Similarly, pregnancy complications, such as anemia, gestational hypertension, pre-eclampsia or eclampsia, and placenta previa or abruptio, were associated with increased use of prenatal antibiotics. Notably, crude data analysis revealed significant differences in the prevalence of ADHD and ASD between infants exposed and unexposed to prenatal antibiotics.

### 3.1. Adjusted Hazard Ratios for ADHD and ASD

The Cox regression models were utilized to adjust covariates, and the results are presented in Table 2. Prenatal antibiotic exposure was found to increase the cumulative incidence of ADHD with a hazard ratio of 1.10 (95% confidence interval (CI): 1.08–1.12). Conversely, prenatal antibiotic exposure had only a borderline effect on the cumulative incidence of ASD with a hazard ratio of 1.04 (95% CI: 0.99–1.09). Cesarean section increased the risk of ADHD but not ASD. Maternal comorbidity with diabetes mellitus, hyperlipidemia, and mental and behavioral disorders was found to increase the risk of developing ADHD. However, only maternal mental and behavioral disorders increased the cumulative incidence of ASD. Among common pregnancy-related complications, gestational diabetes mellitus, gestational hypertension, and pre-eclampsia/eclampsia were found to increase the occurrence of ADHD in childhood. However, maternal anemia had a protective effect on offspring ADHD. Only gestational hypertension increased the occurrence of ASD during the follow-up period. Furthermore, male gender was found to be a risk factor for both ADHD and ASD.

### 3.2. Timing of Prenatal Antibiotic Exposure

In order to investigate the impact of prenatal antibiotic exposure, we conducted further analysis of the timing and dosage of exposure. Table 3 presents a summary of our findings. We found that, for ADHD, exposure to antibiotics during any of three different gestational age ranges significantly increased the cumulative risk. However, only exposure after 34 weeks of gestation had a significant impact on the occurrence of ASD.

### 3.3. The Dose Effects of Prenatal Antibiotic Exposure

Additionally, we explored the dose effects of prenatal antibiotic exposure and summarized our results in Table 4. We found a dose-dependent effect on the occurrence of ADHD but no effect on the occurrence of ASD.

## 4. Discussion

This nationwide population-based study established a positive association between exposure to prenatal antibiotics and the subsequent development of ADHD and ASD during childhood. The study also examined the relevance of the timing of exposure and the dose effect on the development of these conditions. While a dose-dependent effect was observed in ADHD, no such effect was found in ASD. Late preterm and term infants showed the most significant impact of antibiotic exposure on the development of ASD. The study has several strengths, including a large sample size of 906,942 participants and a longitudinal follow-up of at least 6 years. The information collected from our database was comprehensive, and the database enrolled practically all newborns during this period, without discrimination on the basis of social/economic status or region of residence. Moreover, access to health care and medical resources is universal in Taiwan. Thus, diagnoses of ADHD or ASD would not have been overlooked for children in lower social/economic classes. Furthermore, the study employed multiple regression models to control for confounding factors, such as maternal age, type of delivery, maternal comorbidities, and common pregnancy-related complications, and assessed their roles in the development of ADHD/ASD.

Numerous studies have investigated the relationship between early-life exposure and the development of ADHD or ASD. A Canadian population-based study utilized multiple databases to examine the association between antibiotic exposure in the first year of life and the risk of ADHD later on. The study found that exposure to antibiotics during the first year of life was linked to an increased risk of ADHD [19]. However, this study examined postnatal exposure to antibiotics and did not exclude cases of multiple delivery and preterm delivery.

Another twin study was conducted to investigate the association between early-life antibiotic exposure and the risk of developing ADHD and ASD. The findings revealed a positive association between antibiotic use and increased risk of these disorders which was attenuated in monozygotic twins. These results suggest the potential for confounding by a shared familial environment and genetic factors during early development [20]. A systematic review and meta-analysis analyzed the relationship between early-life antibiotic exposure and the development of ASD, concluding that broad-spectrum antibiotics were associated with a higher risk for ASD when compared with penicillin or macrolide antibiotics. This analysis also reported an increased risk of developing ADHD in children who were exposed to antibiotics before the age of 2 years. However, this analysis did not report on prenatal exposure to antibiotics [21].

A cohort study was conducted to investigate the relationship between prenatal antibiotic exposure and the risk of ADHD using a sibling analysis approach. The results showed an increased risk of ADHD in the overall and matched cohorts, but not in the sibling cohort. The authors concluded that there was no significant association between prenatal antibiotic exposure and the risk of ADHD later in childhood. However, the study had a relatively small sample size (*n* = 187,605) compared to our study. Furthermore, this study did not exclude preterm neonates, whose immature brain and nervous systems and prematurity-associated conditions, such as asphyxia and congenital infections, may have influenced the results [22].

In another population-based survey with a relatively smaller number of cases (*n* = 214,834) which examined the association between prenatal antibiotic exposure and the risk of ASD, a small increase in ASD risk was observed following prenatal exposure. The authors also noted the limited clinical significance of this finding, although it was statistically significant and consistent with the results of our study. This study also found higher risk in those exposed to prenatal antibiotics for longer durations, which was compatible with the findings of our study [23].

ASD is a neurodevelopmental condition characterized by impairments in social interaction, social communication, and the presence of restricted and repetitive patterns of behavior, interests, and activities [24]. The prevalence of ASD is increasing in both Western and Eastern nations [25,26,27]. It is currently recognized that approximately one in every hundred children is diagnosed with ASD, and there is a higher incidence in males compared to females, with a four-fold difference in prevalence [28]. The majority of individuals with ASD benefit from ongoing educational support to address their unique learning needs and promote their development [29]. Additionally, ASD patients were reported to have a higher death rate from unnatural causes. It is crucial to provide comprehensive healthcare management and support to address these potential challenges and ensure the safety and well-being of individuals with ASD [30].

Over the past few decades, researchers have dedicated their efforts to understanding the pathomechanism of ASD to identify novel and more effective therapeutic targets [31,32]. While the exact pathogenesis of ASD is not fully understood, there has been a proposal suggesting the involvement of maternal immune activation (MIA) in influencing fetal neurodevelopment through inflammatory and epigenetic mechanisms, ultimately leading to the development of ASD in children. MIA can be triggered by various factors, including genetic mutations, environmental influences, and maternal acute or chronic inflammation. These factors include smoking, obesity, gestational diabetes, pre-eclampsia, depression, psychosocial stress, pollution, low socioeconomic status, asthma, and autoimmune diseases [33]. Our research specifically explores the correlation between prenatal antibiotic usage and ASD, which may be mediated through the mechanism of MIA. In brief, ASD has a significant impact on individuals diagnosed with the condition and their families. Gaining a deeper understanding of the pathogenesis and risk factors associated with ASD can facilitate the discovery of strategies to prevent or reduce the burden of the disease.

Our study had several limitations that require consideration. First, the NHIRD of Taiwan was our data source, resulting in a predominantly Asian cohort. The restricted diversity in ethnicity may have contributed to disparities in our findings compared to analogous studies conducted in Western countries [22,23]. Second, outcome misclassification may have been a potential issue in our database study. The identification of ADHD and ASD was reliant on ICD-9 codes, which may not have encompassed all subtypes of these disorders, thus potentially leading to an underestimation of cases. Despite a follow-up period of at least nine years, we may have missed some children who developed these disorders in adolescence or later. Moreover, while we utilized ATC codes to confirm prenatal exposure to antibiotics and assessed the timings and dose effects of the exposures, we did not further categorize the different classes of antibiotics. Different spectra of antibiotics may exert different effects; therefore, further analysis of the various antibiotic classifications could provide a deeper understanding of their potential impacts. Furthermore, we opted to exclude multiple and preterm deliveries to simplify the study design. Alternative designs, such as twin studies, may be better suited to control for potential confounding factors [20]. Lastly, the analysis of microbiomes, which can be modified by antibiotic exposure, may offer additional insights into their involvement in cognitive and behavioral disorders, such as ADHD and ASD. However, the lack of microbiome data in our database study precludes such an analysis. This study was conducted in Taiwan, an island in East Asia. Most of the study population were oriental people. This may limit its external validity.

## 5. Conclusions

Our findings suggest that prenatal exposure to antibiotics may increase the risk of developing ADHD and ASD later in life. This discovery constitutes an advancement in unraveling the intricate pathogenesis of ADHD and ASD. Furthermore, it shows the potential for identifying preventive strategies specifically designed to mitigate the incidence of ADHD and ASD. However, further prospective follow-up studies are necessary to fully elucidate the association between prenatal antibiotic exposure and the development of these disorders.

## Figures and Tables

**Figure 1 children-10-01128-f001:**
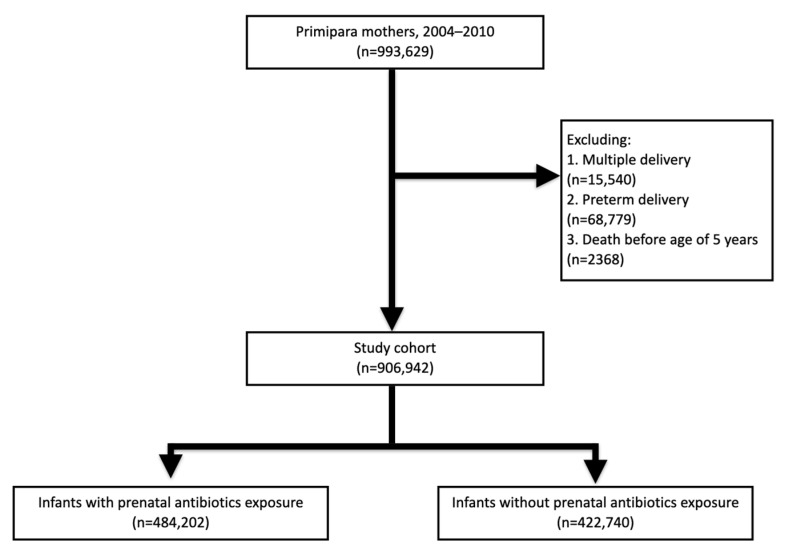
Enrollment of the study cohort.

**Table 1 children-10-01128-t001:** Demographic data of the study cohort.

	No Antibiotics	Antibiotics	Total	*p*-Value
	(*n* = 422,740)	(*n* = 484,202)		
	*n* (%)	*n* (%)		
Maternal age				<0.001
<25	72,366 (17.1)	90,395 (18.7)	162,761	
25–29	159,287 (37.7)	180,320 (37.2)	339,607	
30–34	140,761 (33.3)	151,804 (31.4)	292,565	
≥35	50,326 (11.9)	61,683 (12.7)	112,009	
Mode of delivery				<0.001
Vaginal delivery	313,169 (74.1)	291,785 (60.3)	604,954	
Cesarean section	109,571 (25.9)	192,417 (39.7)	301,988	
Maternal comorbidities				
Diabetes mellitus	1822 (0.4)	3284 (0.7)	5106	<0.001
Hypertension	1752 (0.4)	3452 (0.7)	5204	<0.001
Hyperlipidemia	3365 (0.8)	6074 (1.3)	9439	<0.001
Mental and behavioral disorders	38,092 (9)	65,450 (13.5)	103,542	<0.001
Pregnancy-related complications				
Anemia	15,979 (3.8)	24,690 (5.1)	40,669	<0.001
Gestational diabetes mellitus	6217 (1.5)	7401 (1.5)	13,618	0.024
Gestational hypertension	1485 (0.4)	2389 (0.5)	3874	<0.001
Pre-eclampsia or eclampsia	2986 (0.7)	5390 (1.1)	8376	<0.001
Placenta previa and abruptio placentae	9664 (2.3)	15,581 (3.2)	25,245	<0.001
Neonatal gender				<0.001
Female	205,227 (48.5)	231,737 (47.9)	436,964	
Male	217,513 (51.5)	252,465 (52.1)	469,978	
ADHD				<0.001
No	400,489 (94.7)	455,076 (94)	855,565	
Yes	22,251 (5.3)	29,126 (6)	51,377	
Autism				0.002
No	419,839 (99.3)	480,610 (99.3)	900,449	
Yes	2901 (0.69)	3592 (0.74)	6493	

**Table 2 children-10-01128-t002:** Hazard ratios for attention deficit hyperactivity disorder and autism spectrum disorder in children with or without prenatal antibiotic exposure.

	ADHD				ASD			
	HR	95% CI		*p*-Value	HR	95% CI		*p*-Value
Antibiotics	1.1	1.08	1.12	<0.001	1.04	0.99	1.09	0.12
Maternal age								
<25	1				1			
25–29	0.93	0.91	0.96	<0.001	1.25	1.15	1.35	<0.001
30–34	0.92	0.89	0.94	<0.001	1.82	1.68	1.97	<0.001
≥35	0.97	0.94	1	0.076	2.21	2.02	2.42	<0.001
Mode of delivery								
Vaginal delivery	1				1			
Cesarean section	1.03	1.01	1.05	0.004	1.05	1	1.11	0.08
Maternal comorbidities								
Diabetes mellitus	1.27	1.15	1.4	<0.001	1.26	0.97	1.64	0.09
Hypertension	0.98	0.88	1.09	0.75	0.96	0.72	1.27	0.76
Hyperlipidemia	1.3	1.21	1.4	<0.001	1.16	0.94	1.42	0.17
Mental and behavioral disorders	1.54	1.51	1.58	<0.001	1.55	1.46	1.66	<0.001
Pregnancy-related complications								
Anemia	0.92	0.88	0.96	<0.001	1.01	0.9	1.14	0.86
Gestational diabetes mellitus	1.15	1.08	1.23	<0.001	1.16	0.97	1.38	0.1
Gestational hypertension	1.22	1.08	1.37	0.002	1.38	1.02	1.87	0.038
Pre-eclampsia or eclampsia	1.27	1.17	1.38	<0.001	1.22	0.97	1.52	0.08
Placenta previa and abruptio placentae	1.06	1.01	1.12	0.018	1	0.87	1.15	0.97
Neonatal gender								
Female	1				1			
Male	3.31	3.24	3.38	<0.001	5.46	5.1	5.85	<0.001

ADHD: attention deficit hyperactivity disorder, ASD: autism spectrum disorder, CI: confidence interval, HR: hazard ratio.

**Table 3 children-10-01128-t003:** Timing of antibiotic exposure and risk of attention deficit hyperactivity disorder and autism spectrum disorder *.

	ADHD				ASD			
	HR	95% CI		*p*-Value	HR	95% CI		*p*-Value
Before GA 28 weeks	1.13	1.11	1.16	<0.001	1.01	0.95	1.07	0.776
GA 28 to 34 weeks	1.11	1.06	1.16	<0.001	0.89	0.77	1.03	0.115
After GA 34 weeks	1.05	1.02	1.08	<0.001	1.13	1.05	1.20	<0.001

* Model adjusted for maternal age, mode of delivery, preterm delivery, maternal comorbidities, pregnancy-related complications, and neonatal gender. ADHD: attention deficit hyperactivity disorder, ASD: autism spectrum disorder, CI: confidence interval, HR: hazard ratio, GA: gestational age.

**Table 4 children-10-01128-t004:** Cumulative times of antibiotic exposure during pregnancy and offspring risk for attention deficit hyperactivity disorder and autism spectrum disorder *.

	ADHD				ASD			
	HR	95% CI		*p*-Value	HR	95% CI		*p*-Value
1 time	1.06	1.03	1.08	<0.001	1.05	0.99	1.11	0.130
2 times	1.13	1.10	1.16	<0.001	1.03	0.95	1.11	0.491
≥3 times	1.21	1.18	1.25	<0.001	1.04	0.96	1.13	0.343

* Model adjusted for maternal age, mode of delivery, preterm delivery, maternal comorbidities, pregnancy-related complications, and neonatal gender. ADHD: attention deficit hyperactivity disorder, ASD: autism spectrum disorder, CI: confidence interval, HR: hazard ratio.

## Data Availability

The datasets presented in this article are not readily available because data release is not allowed by the National Health Insurance Research Database. Requests to access the datasets should be directed to Ching-Heng Lin (epid@vghtc.gov.tw).

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
