# Peer review of "The Association of Prenatal Antibiotic Use with Attention Deficit and Autism Spectrum Disorders: A Nationwide Cohort Study"

_children, 2023, doi:10.3390/children10071128_

Round 1

Reviewer 1 Report

The article entitled "Associations of prenatal antibiotics use with attention deficit and autism spectrum disorders" was considered for review. The authors aimed to examine the potential association between prenatal exposure to antibiotics and the risk of ADHD and ASD in childhood from a nation-wide perspective. The article is relevant to the pediatric field. However, some specific modifications are needed to improve the paper, which are described below:

Title: Please indicate the type of study.

Abstract: The objective presented (lines 15 and 16) does not align with the one presented in the Introduction section (lines 57 to 59), which highlights that the analysis will focus on childhood. Therefore, it needs to be adjusted accordingly.

Introduction: I suggest removing the reference to the problematization of the local or systemic microbiome and the development of disorders such as ADHD and ASD, as it was not addressed in the results of this study. The writing implies to the reader that it will be examined. I suggest replacing the term "psychiatric disorder" (line 54). Please include any pre-existing hypotheses.

Methods: There are weaknesses that need clarification. The authors provide an adequate explanation of the eligibility criteria, sources, and methods of participant selection. However, a better description of the methods for monitoring the included children is necessary. They mention that the cohort was followed until the end of 2016 and that all children were followed for at least 6 years (lines 87 and 88), but they do not explain how this follow-up was conducted. The cohort flow diagram also needs to be reviewed as the sum is incorrect (presumably due to the exclusion of data related to item 3 of the exclusion criteria: Death before age 5 years). If applicable, please explain how loss to follow-up was handled.

Results: I suggest moving the flow diagram to the beginning of this section and including the number of participants who completed the follow-up and were effectively analyzed. Please provide the follow-up period (e.g., mean and total time). Describe the reasons for losses at each stage.

Discussion: It is well-written, presenting similarities to other authors but emphasizing the novel aspects of the study. The author's interpretations demonstrate confidence and expertise, advancing knowledge and allowing for replication. Limitations were also satisfactorily presented. I suggest improving the discussion regarding the generalizability (external validity) of the results.

Conclusions: No recommendations.

References are appropriate in number and up to date.

Author Response

To Reviewer #1:

Thank you for your comments and suggestions. The questions are answered below:

1. Re: Title: Please indicate the type of study.

Ans: The study type has been added to the tile. (Line 3)

2. Re: Abstract: The objective presented (lines 15 and 16) does not align with the one presented in the Introduction section (lines 57 to 59), which highlights that the analysis will focus on childhood. Therefore, it needs to be adjusted accordingly.

Ans: The abstract and introduction sections have been adjusted accordingly. (Line 16-17)

3. Re: Table 1. Characteristics of study subjects revealed many factors eq. Family income, Urbanization, Mode of delivery, Pregnancy-related complication, Birth weight were significant different between control group and GDM group. The authors should explain the difference.

Ans: The content of table 1 has been described between as “Advanced or younger maternal age was positively associated with increased antenatal antibiotic use, as was delivery by cesarean section. Maternal comorbidities such as dia-betes, hypertension, hyperlipidemia, and mental health issues were also found to be associated with an increased likelihood of antenatal antibiotic exposure. Similarly, pregnancy complications such as anemia, gestational hypertension, pre-eclampsia or eclampsia, and placenta previa or abruptio were associated with increased use of prenatal antibiotics. Notably, crude data analysis revealed significant differences in the prevalence of ADHD and ASD between infants exposed and unexposed to prenatal antibiotics.” (Line 129-137)

4. Re: Introduction: I suggest removing the reference to the problematization of the local or systemic microbiome and the development of disorders such as ADHD and ASD, as it was not addressed in the results of this study. The writing implies to the reader that it will be examined. I suggest replacing the term "psychiatric disorder" (line 54). Please include any pre-existing hypotheses.is.

Ans: The “psychiatric disorder” has been removed. (Line 56-58) Thank you for your suggestion about removing references. However, gut-brain axis is the core hypothesis of this article. It is an important rationale for conducting this research. We would like to keep this section in the introduction section. We have added some explanations in the text. (Line 53-55)

5. Re: Methods: There are weaknesses that need clarification. The authors provide an adequate explanation of the eligibility criteria, sources, and methods of participant selection. However, a better description of the methods for monitoring the included children is necessary. They mention that the cohort was followed until the end of 2016 and that all children were followed for at least 6 years (lines 87 and 88), but they do not explain how this follow-up was conducted. The cohort flow diagram also needs to be reviewed as the sum is incorrect (presumably due to the exclusion of data related to item 3 of the exclusion criteria: Death before age 5 years). If applicable, please explain how loss to follow-up was handled.

Ans: The cohort flow diagram also needs to be reviewed and corrected. (Figure 1) A total of 906,942 infants were enrolled in the study cohort after applying the exclusion criteria. This nationwide cohort study included only first-time pregnancies from 2004 to 2010. The cohort was followed up until the end of 2016. All children were followed up for at least 6 years. This is a claim data study. Only children with claim data during the study period were included. Children who died during the study period have been excluded. The follow up time is 6 to 12 years. The mean follow-up period is around 9 years. (Line 118-124)

6. Re: Results: I suggest moving the flow diagram to the beginning of this section and including the number of participants who completed the follow-up and were effectively analyzed. Please provide the follow-up period (e.g., mean and total time). Describe the reasons for losses at each stage.

Ans: The diagram has been moved to the result section.  A total of 906,942 infants were enrolled in the study cohort after applying the exclusion criteria. This nationwide cohort study included only first-time pregnancies from 2004 to 2010. The cohort was followed up until the end of 2016. All children were followed up for at least 6 years. This is a claim data study. Only children with claim data during the study period were included. Children who died during the study period have been excluded. The follow up time is 6 to 12 years. The mean follow-up period is around 9 years. (Line 118-124)

7. Re: Discussion: It is well-written, presenting similarities to other authors but emphasizing the novel aspects of the study. The author's interpretations demonstrate confidence and expertise, advancing knowledge and allowing for replication. Limitations were also satisfactorily presented. I suggest improving the discussion regarding the generalizability (external validity) of the results.

Ans: This study is conducted in Taiwan, an island at east Asia. Most of the study population were oriental people. Most of the included subjects were oriental people. This may limit its external validity. The description about external validity has been added at Line 276-278.

Reviewer 2 Report

I read the paper but the original version has just a few improvements.

Also, the conclusion section is too short and does not support the paper. Please try to emphasize your contribution to the field and try to offer a more pragmatic approach to the subject.

I will be waiting for a new, revised version.

The English is fine

Author Response

To Reviewer #2:

Thank you for your comments and suggestions. The questions are answered below:

1. Re: I read the paper but the original version has just a few improvements.

Ans: We have made further enhancements to our manuscript regarding our methodology, results, and discussion, in order to provide a more comprehensive understanding of our research. We are committed to further improving the quality of our work and would greatly appreciate any suggestions or feedback you may have.

2. Re: the conclusion section is too short and does not support the paper. Please try to emphasize your contribution to the field and try to offer a more pragmatic approach to the subject.

Ans: We have added the following paragraph in the discussion section “ASD is a neurodevelopmental condition characterized by impairments in social interaction, social communication, and the presence of restricted and repetitive patterns of behavior, interests, and activities. The prevalence of ASD is increasing in both western and eastern nations, with estimated rates on the rise. It is currently recognized that approximately one in every hundred children is diagnosed with ASD, and there is a higher incidence in males compared to females, with a four-fold difference in prevalence. The majority of individuals with ASD require ongoing educational support. Additionally, ASD has been associated with a significantly increased risk of mortality. Over the past few decades, researchers have dedicated their efforts to understanding the pathomechanism of ASD to identify novel and more effective therapeutic targets. While the exact pathogenesis of ASD is not fully understood, there has been a proposal suggesting the involvement of maternal immune activation (MIA) in influencing fetal neurodevelopment through inflammatory and epigenetic mechanisms, ultimately leading to the development of ASD in children. MIA can be triggered by various factors including genetic mutations, environmental influences, and maternal acute or chronic inflammation. These factors encompass aspects such as smoking, obesity, gestational diabetes, pre-eclampsia, depression, psychosocial stress, pollution, low socioeconomic status, asthma, and autoimmune diseases. Our research specifically explores the correlation between prenatal antibiotics usage and ASD, which may be mediated through the mechanism of MIA. In summary, ASD has a significant impact on individuals diagnosed with the condition and their families. Gaining a deeper understanding of the pathogenesis and risk factors associated with ASD can facilitate the discovery of strategies to prevent or reduce the burden of the disease.” (Line 232-257)

Round 2

Reviewer 1 Report

Editora Dera, O artigo intitulado "Associações do uso de antibióticos pré-natal com déficit de atenção e transtornos do espectro do autismo" foi considerado para revisão. Os autores tiveram como objetivo examinar a associação potencial entre a exposição pré-natal a antibióticos e o risco de TDAH e TEA na infância de uma perspectiva nacional.

Os autores atenderam às transferências feitas e justificaram não retirar a referência à problematização do microbioma local ou sistêmico e ao desenvolvimento de transtornos como TDAH e TEA, ainda pouco compreendidos (linhas 54 e 55).

Agradeço a oportunidade de ter revisado este artigo e recomendo aceitar o manuscrito em sua forma atual.

Author Response

Thank you for your comments and suggestions. The questions are answered below:

1.Re: Editora Dera, O artigo intitulado "Associações do uso de antibióticos pré-natal com déficit de atenção e transtornos do espectro do autismo" foi considerado para revisão. Os autores tiveram como objetivo examinar a associação potencial entre a exposição pré-natal a antibióticos e o risco de TDAH e TEA na infância de uma perspectiva nacional. Os autores atenderam às transferências feitas e justificaram não retirar a referência à problematização do microbioma local ou sistêmico e ao desenvolvimento de transtornos como TDAH e TEA, ainda pouco compreendidos (linhas 54 e 55). Agradeço a oportunidade de ter revisado este artigo e recomendo aceitar o manuscrito em sua forma atual..

Ans: Thank you for your time and effort to review our article.

Reviewer 2 Report

There is some improvement in the paper. The authors took into consideration most of the reviewer's indications.

The present form of the papers has brought new and detailed info.

I still consider that the Conclusion section should be improved and the contribution of the authors to the field should be highlighted in the Conclusions section. 

English is fine

Author Response

Thank you for your comments and suggestions. The questions are answered below:

1. Re: I still consider that the Conclusion section should be improved and the contribution of the authors to the field should be highlighted in the Conclusions section.

Ans: We have added the following description in the conclusion section: “This discovery provides an advancement in unraveling the intricate pathogenesis of ADHD and ASD.  Furthermore, it possesses potential in identifying preventive strategies specifically designed to mitigate the incidence of ADHD and ASD.”